# Barriers in the Nervous System: Challenges and Opportunities for Novel Biomarkers in Amyotrophic Lateral Sclerosis

**DOI:** 10.3390/cells14110848

**Published:** 2025-06-05

**Authors:** Lorena Pisoni, Luisa Donini, Paola Gagni, Maria Pennuto, Antonia Ratti, Federico Verde, Nicola Ticozzi, Jessica Mandrioli, Andrea Calvo, Manuela Basso

**Affiliations:** 1Department of Cellular, Computational and Integrative Biology-CIBIO, University of Trento, 38123 Trento, Italy; luisa.donini@unitn.it; 2Consiglio Nazionale delle Ricerche, Istituto di Scienze e Tecnologie Chimiche “Giulio Natta” (SCITEC-CNR), 20131 Milano, Italy; paola.gagni@cnr.it; 3Department of Biomedical Sciences, University of Padova, 35131 Padova, Italy; maria.pennuto@unipd.it; 4Veneto Institute of Molecular Medicine (VIMM), 35129 Padova, Italy; 5Department of Medical Biotechnology and Translational Medicine, Università degli Studi di Milano, 20133 Milan, Italy; antonia.ratti@unimi.it; 6Department of Neurology and Laboratory of Neuroscience, IRCCS Istituto Auxologico Italiano, 20145 Milan, Italy; f.verde@auxologico.it (F.V.); nicola.ticozzi@unimi.it (N.T.); 7Department of Pathophysiology and Transplantation, “Dino Ferrari” Center, Università degli Studi di Milano, 20122 Milan, Italy; 8Department of Neurosciences, Ospedale Civile Baggiovara, Azienda Ospedaliero Universitaria di Modena, 41126 Modena, Italy; jessica.mandrioli@unimore.it; 9Department of Biomedical, Metabolic and Neural Sciences, University of Modena and Reggio Emilia, 41125 Modena, Italy; 10ALS Centre, “Rita Levi Montalcini” Department of Neuroscience, University of Torino, 10120 Torino, Italy; andrea.calvo@unito.it; 11Azienda Ospedaliero-Universitaria Città della Salute e della Scienza di Torino, SC Neurologia 1U, 10120 Torino, Italy

**Keywords:** blood–CSF barrier, blood–brain barrier, blood–spinal cord barrier, review

## Abstract

Amyotrophic Lateral Sclerosis (ALS) is a complex neurodegenerative disorder characterized by wide phenotypic heterogeneity. Despite efforts to carefully define and stratify ALS patients according to their clinical and genetic features, prognosis prediction still remains unreliable. Biomarkers that reflect changes in the central nervous system would be useful, but the physical impossibility of direct sampling and analysis of the nervous system makes them challenging to validate. A highly explored option is the identification of neuronal-specific markers that could be analyzed in peripheral biofluids. This review focuses on the description of the physical and biological barriers to the central nervous system and of the composition of biofluids in which ALS disease biomarkers are actively searched. Finally, we comment on already validated biomarkers, such as the neurofilament light chain, and show the potential of extracellular vesicles (EVs) and cell-free DNA as additional biomarkers for disease prediction.

## 1. Barriers Delimiting and Protecting the Nervous System

In the central nervous system (CNS), we can identify three main barriers regulating the influx of solvents and preventing the intravenous influx of neurotoxic and vasculogenic molecules that may harm CNS cells [1]. These barriers are represented by the blood–cerebrospinal fluid barrier (BCSFB), the blood–brain barrier (BBB), and the blood–spinal cord barrier (BSCB). There is also an extra-neuronal barrier, called the gut–vascular barrier (GVB), whose activity also affects the CNS [2] (Figure 1).

### 1.1. The Blood–CSF Barrier (BCSFB)

The BCSFB is represented by the choroid plexus located in each of the four cerebral ventricles. Remarkably, the choroid plexus is made of meningeal granular protrusions in the lumen of the cerebral ventricles. These are characterized by a central fenestrated capillary, derived from anterior and posterior choroidal arteries, and an epithelial layer of choroidal cells, which are continuous with the ependymal cells covering the cerebral ventricles’ surface [3]. The choroidal cells are polarized with a basolateral side in contact with the blood and an apical side characterized by villi releasing the cerebrospinal fluid (CSF) into the cerebral ventricles [4]. These cells are connected with tight and adherens junctions that allow the selective passage of ions and other micronutrients, such as vitamins B1, B12, and C, preventing the paracellular movement of molecules [3,5].

### 1.2. The Blood–Brain Barrier (BBB)

Brain vascularization derives from large and interconnected arteries forming the circle of Willis at the base of the brain, an anastomotic arterial network with a polygonal shape that provides blood nutrient supply to the CNS [6,7].

At the cellular level, endothelial cells, pericytes, vascular smooth muscle cells, astrocytes, oligodendrocytes, and neurons constitute the neurovascular unit (NUV), essential to the nervous system’s functionality. Particularly, endothelial cells, pericytes, and astrocytes are involved in forming the BBB at the level of the vast capillary network which vascularizes the brain. The BBB has the important role of controlling the exchange of molecules between the blood and the CNS, preventing the passage of potentially toxic molecules [8,9,10].

Endothelial cells are enriched with mitochondria and express proteins involved in forming tight and adherens junctions (claudins, occludins, junctional adhesion molecules, and Zonula Occludens (ZO)) [11]. The cell junctions are essential for providing a physical barrier, allowing the free passage of only oxygen, carbon dioxide, and small lipophilic molecules from the blood to the interstitial fluid to prevent the influx of potentially toxic molecules that could damage the CNS. These cells also provide a biochemical barrier through the expression of many transporters of efflux, which pump out unwanted molecules, as well as carriers that regulate the trans-endothelial movement of carbohydrates, hormones, ions, and other molecules circulating into the bloodstream [10]. Concerning the biochemical barrier properties of the BBB, several ATP-binding cassettes, the breast cancer resistance protein (BCRP), and multidrug-resistance-associated proteins (MRPs) are expressed, with the most important ones represented by P-glycoprotein and ATP-binding cassette subfamily G member 2 (ABCG2) [12]. Moreover, the endothelial cells of the BBB express some enzymes of the cytochrome P450 family, with cytochrome P450 family 1 subfamily B (CYP1B1) being the predominant one [12].

The pericytes form direct contact with the endothelium through N-cadherin and connexins, encapsulating the capillary wall [8]. These cells regulate the integrity of the BBB and the permeability of the blood vessels. In various transgenic mouse models expressing mutant platelet-derived growth factor (PDGF)-β/PDGF receptor β signaling, which is involved in the recruitment of pericytes during angiogenesis, proliferation, and migration, several tracers of different molecular weights accumulated in the brain parenchyma, unlike in wild-type mice, showing the loss of integrity of the BBB [13]. Pericytes also have contractile features similar to smooth muscle cells, thus allowing them to regulate the diameter of the brain vessels and the flux of blood into the brain and retinal capillaries in response to vasoactive molecules such as catecholamines, endothelin-1, vasopressin, angiotensin II, ATP, and glutamate [14,15]. In support of this, a reduction in the pericyte population is correlated with an increased vessel diameter and vessel density.

Finally, the astrocytes, with their endfoot processes, define the perivascular space, regulate the ions’ homeostasis, and control the flux of water into the parenchyma through the expression of several transporters and channels. These cells are highly polarized and express protein carriers at the interface with the blood vessels [16,17]. An astrocyte-specific carrier is represented by aquaporin 4 (AQP4), whose expression is restricted to astrocytes and to a subpopulation of ependymal cells localized in the subfornical organ and the supraoptic nucleus. AQP4 is implicated in the osmoregulation between brain parenchyma, CSF, blood, and the clearance of CNS [18,19,20].

### 1.3. The Blood–Spinal Cord Barrier (BSCB)

Similarly to the BBB, the blood vessels that provide nutrients to the spinal cord are highly specialized to regulate the exchange of solutes between blood and the spinal cord parenchyma. The BSCB is morphologically similar to the BBB and hence characterized by non-fenestrated capillaries with endothelial cells strictly associated with tight and adherens junctions, which prevent the transcellular diffusion of molecules, and expressing influx and efflux transporters that finely control the transcellular transport. Endothelial cells are closely associated with pericytes and astrocytes, and their cytoplasmic endfeet envelop the blood vessels [21,22].

This barrier is more permeable than the BBB; in fact, the expression of tight junction proteins, such as ZO-1, occludin, vascular endothelial cadherin, and β-catenin, is reduced in BSCB [23]. Consistent with this observation, Winkler et al. demonstrated that, upon injection of fluorescent cadaverine, this biogenic amine accumulates in the spinal cord, but not in the brain parenchyma [24]. Furthermore, the immunostaining of CD13 and PDGFRB revealed a lower pericyte coverage in the anterior horn capillaries of the cervical, thoracic, and lumbar spinal cord compared to the brain capillaries. This condition leads to an increased paracellular flow of molecules, favoring the accumulation of plasma proteins in the spinal cord [24].

### 1.4. Defects in the CNS Barriers in Amyotrophic Lateral Sclerosis

Functional alterations of cell types within the BBB, such as endothelial or mural cells, cause vascular defects and neurological deficits [25]. Changes in the structure of the CNS barriers have been reported in neurodegenerative diseases, such as Alzheimer’s disease (AD), Parkinson’s disease (PD), Huntington’s disease (HD), and Amyotrophic Lateral Sclerosis (ALS), as well as in acute neurological disorders [26].

ALS is the most common motor neuron disease, characterized by the selective degeneration of upper and lower motor neurons [27]. ALS is extremely heterogeneous clinically and genetically [28]. A total of 90% of individuals with ALS present a sporadic form, while in 10% of cases, the disease is familial, with more than 30 genes considered pathogenetic. Accumulation of an RNA-binding protein known as TAR DNA-binding protein 43 (TDP-43), encoded by the gene TARDPB, has been reported in 97% of ALS cases [27]. Structural and functional abnormalities of the barriers that protect the CNS have been demonstrated in animal models mimicking ALS pathology and in human postmortem brain tissues, in studies with a prominent focus on the BSCB; only a few studies have also assessed the BBB in the cortex in ALS [29].

Several mouse and rat experimental models, comprising mice carrying *SOD1* and *TARDBP* mutations, along with *C9ORF72* repeat expansion, have been used to study alterations of the CNS barriers from a presymptomatic to a late stage of the disease. In particular, leakage of microvessels, possibly caused by endothelial and basement membrane dysfunction, showed diffusion of Evans Blue, a dye that binds to serum albumin, in the parenchyma of the cervical and lumbar spinal cord from transgenic mutant SOD1 rats and mice at the early symptomatic stage [30,31], along with deposits of hemosiderin, a breakdown product of hemoglobin, in CNS parenchyma [32]. The tight junction proteins ZO-1, occludin, and claudin-5 were found to have reduced expression in spinal cord capillaries from presymptomatic mutant SOD1 mouse models in comparison with control littermates; this is an event happening before motor neuron degeneration which is then exacerbated at the end stage of the disease [33,34]. ZO-1 reduction was confirmed in cortical endothelial cells of the brain microvasculature in transgenic C9orf72 ALS mice at the early stage of the disease [35]. Alterations in the basement membrane in transgenic SOD1^G93A^ mice have also been reported; in particular, collagen IV is reduced in the vascular structure, predominantly in the anterior horn [36], and laminin-1 is decreased, suggesting basement membrane disruption [30]. Moreover, in ALS rats, a decreased expression of agrin, a proteoglycan component of the basement membrane, is observed in association with the disorganization of the extracellular matrix [31]. AQP4, mainly localized in the astrocytic endfeet, is also increased in transgenic SOD1^G93A^ mice [37,38].

Another aspect concerns the induction of perivascular fibroblasts at the presymptomatic stage, possibly involved in remodeling blood vessels in the CNS. In transgenic SOD1^G93A^ ALS mice, there was an increased expression of collagen type VI alpha 1 chain (COL6A1) and secreted phosphoprotein 1 (SPP1) [39]. This was paired with a reduction in capillary density and blood flow in the anterior horn of the lumbar cord occurring at the presymptomatic stage [32,36]. Furthermore, in an in vitro model of BBB from patient donors carrying C9orf72 expansion, compromised barrier integrity and increased activity of the P-glycoprotein transporter have been recently reported [40]. No brain vascular leakage or altered BBB passive diffusion was observed in C9orf72 mice [35], and no ultrastructural changes of BSCB in the brainstem, cervical, and lumbar spinal cords were revealed in capillary inter-endothelial tight junctions in SOD1^G93A^ mice [41], suggesting a discrepancy between protein alteration, functional, and structural measurements.

Most of the observations in murine ALS models have been confirmed in human postmortem ALS specimens. Several independent neuropathological studies revealed CNS barrier perturbation and changes in the neurovascular unit composition in postmortem tissues collected from sporadic ALS patients. At the level of the medulla, spinal cord, and choroid plexus, the protein levels of ZO-1, occludin, claudin-1-3-5, junctional adhesion molecule 1 (JAM-1), and vascular endothelial (VE)-cadherin are reduced [42,43,44]. In BSCB and BCSFB, the staining of capillary endothelium with PECAM-1 and CD105 markers emphasizes a discontinuous endothelial lining [42] along with a reduction in the number of PDGFRβ-positive cells and erythrocyte extravasation in cervical spinal cord anterior horn gray matter [45]. Similarly, Yamadera et al. found a significant decrease in pericyte coverage in the ventral horn of ALS patients compared to controls; this abnormality was also linked to a significantly increased microvascular density [46]. The pericyte reduction was also confirmed at the level of the BCSFB in the choroid plexus [44]. Like in animal models, disorganized collagen IV accumulates in the surrounding leaked vessels. Microvascular barrier abnormalities have been observed in diseased tissues, with microvascular remodeling in the gray and white matter of patients’ medulla, cervical, and lumbar spinal cords. Microvessels are deformed in the spinal cord of ALS patients, with an increased density in patients who underwent artificial respiratory support [46].

Nevertheless, it is still unclear whether the alterations of CNS barriers are an early event contributing to disease onset or just a downstream event exacerbating motor neuron degeneration during disease progression [45]. The analysis of biofluids collected from ALS patients can certainly provide some insights to address this issue. In fact, blood-derived proteins have been identified in the CSF [47,48], and several CNS-specific proteins, like neurofilaments, glial fibrillary acidic protein (GFAP), and tau, are currently measured in the blood of ALS patients, thus offering new means to monitor neuronal health.

### 1.5. Extra-Neuronal Barriers Affecting the CNS: The Gut–Vascular Barrier

Recent evidence demonstrates a direct link between the CNS and the gut. The gut has a physical and immunological barrier that prevents microorganisms and toxic substances from entering the body. This physical semipermeable barrier is composed of a monolayer of epithelial cells that are strictly associated with each other, thanks to tight and adherens junctions, preventing paracellular flow. Interspersed between epithelial cells, goblet cells secrete mucus, protecting the epithelium from toxic materials and pathogens. At the level of the gut’s blood vessels, a physical barrier that shares several features with the BBB is called the gut–vascular barrier (GVB). Starting from the vessel’s lumen, endothelial cells, pericytes, and enteric glial cells prevent the paracellular flow of molecules bigger than 4kDa [49]. Enteric glial cells (EGCs) play a key role in maintaining the barrier’s integrity [50]. Accordingly, mice with induced ablation of EGC show higher barrier permeability, the release of substances and bacteria into the bloodstream, and a severe inflammatory response [51].

The gut microbiota includes several species of the realms of bacteria, fungi, and viruses, most of which are commensal or mutualistic microorganisms. The composition varies from person to person, depending on several factors such as demography, ethnicity, sex, age, diet, exposure to antimicrobial agents, and healthy or pathological state [52,53,54].

The microbiota has been demonstrated to be involved in several functions of the organism, including the regulation of CNS development and homeostasis [55]. Among these functions, the gut residents regulate the turnover of neurotransmitters, such as noradrenalin, dopamine, and serotonin, and the expression of several proteins associated with synaptogenesis, including postsynaptic density protein 95 (PSD95) and synaptophysin in the striatum. Furthermore, the composition of the microbiota also impacts behavior and anxiety [54,56,57].

The composition of gut microbiota species is essential to benefit the host, including the BBB integrity. In support of this, mice born from germ-free mothers show higher BBB permeability without presenting differences in vascularization and pericyte coverage. Decreased expression of tight junction proteins, like claudin 5, ZO-1, and occludin, in the striatum, cortex, and hippocampus is re-established by colonizing the germ-free mice with the gut microbiota of control mice [58].

Alterations in the composition of the microbiota have been reported in mouse models and patients of several chronic diseases, such as metabolic disorders, atherosclerosis, asthma, autism spectrum disorder, and neurodegenerative disorders [59]. In neurodegenerative disorders, gut microbiota alterations are associated with a dysfunction of the BBB [59]. In an in vivo model of multiple sclerosis (MS), claudin 5 is decreased, and the BBB is more permeable [60]. In animal models of MS, fecal microbiota transplantation (FMT) is associated with a decrease in axonal damage markers (e.g., myelin binding protein and neurofilament light polypeptide (NFL)), intact myelin sheaths in the thoracic spinal cord, and a reduction in the number of activated microglia [61]. Also, germ-free mice show a decrease in the activation of microglia, motor impairment, and inclusion of alpha-synuclein after both supplementation of short-chain fatty acids (SCFAs) and FMT from healthy human donors; inversely, when an FMT is performed from PD patients, the symptoms are exacerbated [62]. The absence of gut microbiota influences the integrity of both BBB and BCSFB by decreasing tight junction expression. This phenotype is rescued in germ-free mice and mice treated with antibiotics after recolonization and in broad-spectrum antibiotic-treated mice by supplementary SCFAs such as propionate and butyrate [63].

This practice is now being explored in humans, and it has been proven to be well tolerated and safe, although efficacy data are still lacking [64,65]. Two ALS patients who underwent FMT showed improvement on the clinical scale ALSFRS-R [59]. A recent Chinese trial was unable to demonstrate an effect of FMT on ALSFRS-R, but recruitment was terminated early before reaching the prespecified sample size due to funding constraints [66]. The results of a recently terminated clinical trial of FMT in 42 ALS patients in Italy are awaited [67]. Notably, individuals with ALS and spinal onset presented gastrointestinal dysbiosis, while patients with a bulbar form of ALS showed dysbiosis in the saliva microbiota [68]. Deciphering how the general microbiome and the GVB influence the integrity of the BBB, BSCB, and BCSFB in ALS and identifying biomarkers of these events could be an added value to monitoring the disease progression.

## 2. Does the Damage to the Barriers Offer Opportunities to Search for Biomarkers in Circulation?

The physical rupture or damage of cellular barriers should favor the exchange of tissue-specific molecules between different body districts and the search for altered circulating biomarkers in the different biofluids. Indeed, detecting differentially expressed molecules in biofluids in a disease state compared to the physiological condition is promising and receiving great attention in ALS research. However, it is not trivial to detect CNS-specific proteins in peripheral biofluids.

Here, we will give an overview of the main features of the different human biofluids, focusing on well-established CNS-protein biomarkers for ALS disease and illustrating novel avenues of investigation (Figure 2).

### Biofluids for Biomarker Detection

The cerebrospinal fluid and the interstitial fluid

The cerebrospinal fluid (CSF) is a chemically stable biofluid that circulates in the CNS, particularly in the subarachnoid space and cerebral ventricles [69]. It has several functions, including regulating the homeostasis of the interstitial fluid of the brain parenchyma, reducing the weight of the CNS by 30 times through its buoyancy property, nourishing and protecting the CNS from mechanical injury, and removing metabolic waste. A human adult contains almost 150 milliliters of CSF, which is completely replaced four to five times every 24 h. This replacement rate is reduced in the elderly [70]. CSF is not a simple blood filtrate; it exhibits a higher concentration of sodium, chloride, folate, amino acids, and magnesium and a lower concentration of potassium, phosphate, albumin, glucose, and calcium than blood [71]. The protein concentration is low, with only 0.025 g of protein per 100 mL, the majority of which is albumin, and the number of cells is less than 5 per mL of CSF. Ions and micronutrients like glucose, vitamins, folate, vasopressin, nitric oxide, and arginine pass through the epithelial cells into the cerebral ventricles at the level of the choroid plexus. This process leads to an increase in the osmotic gradient and, consequently, the passage of water into the CSF through aquaporin 1 [70].

More specifically, the CSF is produced at the level of the choroid plexus, which is in each cerebral ventricle, and at the level of the capillaries of the BBB. The production of CSF is regulated by different factors, such as intraventricular pressure, the autonomic nervous system, Atrial Natriuretic Peptide (ANP), vasopressin, and others. The CSF enters the subarachnoid space through the foramen of Magendie, located in the fourth ventricle, which surrounds the brain and the spinal cord. Next, the CSF can be reabsorbed at the level of the arachnoid granulations or flow into the cranial and spinal subarachnoid space and, thus, into the Virchow–Robin perivascular space. In particular, the subarachnoid space is continuous with the perivascular space surrounding the cortical and penetrating arteries [72]. Iliff et al., in 2012, demonstrated that the CSF flows between the media tunica and the astrocytic endfeet of the arteries and veins driven by arterial pulsation [16]. Based on their size, the solutes present in the CSF can pass through the endfeet of the astrocytes and enter the parenchyma. Here, the bulk flow, which depends upon the movement of water across the astrocytes, determines the flow of the interstitial fluid into the perivenous space and so the clearance of the parenchyma from the metabolic waste, including, for instance, amyloid β 1–40 [16]. This process is associated with the so-called “glymphatic system”, named for its lymphatic-like function and because it depends on glial water flux mediated by AQP4 expressed in the endfeet of astrocytes (https:\doi.org\10.1016\j.nbd.2023.106035). The interstitial fluid enriched with the cells’ metabolic waste must be drained into the bloodstream; however, how the drainage of waste and liquid occurs in the CNS is not as well characterized as in the periphery. One important CSF efflux route is represented by the basal and dorsal meningeal lymphatic vessels. They are lymphatic vessels located in the dura mater meningeal sheath, deprived of the smooth muscle layer and with limited lymphatic valves [73,74,75]. These vessels are mainly localized around the venous sinuses, and with a higher density in the meningeal than in the periosteal layer of the dura mater [76]. The involvement of the meningeal lymphatic vessels in liquor reabsorption was discovered in 2015 in mouse models. Both Louveau and Aspelund demonstrated with fluorescent tracers that the interstitial fluid and the CSF are drained into the subarachnoid space, then into the meningeal lymphatic, and subsequently move into the deep cervical lymph nodes, and then into the superficial cervical lymph nodes [73,74]. In support of this, a reduced clearance and a subsequent accumulation of ovalbumin in deep cranial lymph nodes are reported in transgenic mice defective for lymphatic vessels. Recently, amyloid β 1–40, phosphorylated tau 181 (pTau181), GFAP, and NFL were successfully measured in the human cervical lymph node, providing an additional compartment to analyze and monitor neurodegenerative processes [77].

Due to its proximity to the neuronal and glial cells constituting the nervous system, the CSF is considered the best source for exploring biomarkers for neurological conditions. The most reproducible biomarkers for ALS that have been identified in the CSF are neuronal structural proteins such as NFL, neurofilament heavy (NFH and phosphorylated NFH), tau, and its phosphorylated forms (phosphorylated Tau 181 and 217) [78]. In addition, there are also chitinases, enzymes secreted by macrophage and activated microglia in the inflammatory state, namely the myeloid protein chitotriosidase-1 (CHIT1), the glial protein YKL-40, also known as chitinase-3-like protein 1 (CHI3L1), and YKL-39, also known as chitinase-3-like protein 2 (CHI3L2) (reviewed in [79]). Specifically, each of these markers could have different applications in clinical practice. Neurofilaments have been identified as the most promising biomarkers for ALS, having diagnostic, prognostic, and disease monitoring roles [80]. Chitinases demonstrate diagnostic value, with CHIT1 and CHIT3L1 being associated with disease severity and progression [81]. Instead, tau and its phosphorylated forms correlate with disease severity [82]. Of interest, a novel diagnostic protein biomarker is derived from the cryptic exon-containing hepatoma-derived growth factor-like protein 2 (HDGFL2), whose content in the CSF is strictly dependent on TDP-43 loss of splicing activity in ALS and FTD and is significantly increased in sporadic ALS patients as well as in presymptomatic and symptomatic *C9ORF72* mutation carriers [83]. Unfortunately, the CSF collection is an invasive intervention with a lumbar puncture [84]. Therefore, the applicability of these biomarkers is limited and not for routine monitoring. The most studied CSF biomarkers are reported in Table 1.

The blood

A human adult contains approximately 4–5 L of blood [85]. This liquid tissue is composed of a cellular fraction (45%) and a liquid phase called plasma (55%). The blood has several functions, including the transport of oxygen and carbon dioxide from the lungs to the tissues and vice versa, and the transport of metabolites, ions, and nutrients [86]. It is involved in the hemostasis process and in protecting the organism from infections. It also carries substances from the site of production or storage to target organs, such as hormones and vitamins [87]. More specifically, the corpuscular part of the blood comprises red blood cells, white blood cells, and platelets. These cells are produced starting from hemopoietic stem cells (HSCs) in the bone marrow, specifically at the level of the HSC niches [88].

Plasma mainly comprises water (90–92%) and more than a thousand proteins, electrolytes, nutrients, metabolites, and dissolved gases (O_2_, CO_2_, and N_2_). The most abundant proteins are albumin, globulins, and fibrinogen [89]. The majority of plasma proteins, such as albumin, fibrinogen, and the other coagulation factors, are produced at the level of the liver, whereas immunoglobulins are produced by B cells [85].

Blood is highly investigated due to its ease of accessibility and low invasiveness. However, it is an extremely complex matrix to analyze, due to the high level of soluble proteins and lipoproteins that could influence the identification of biomarkers [79]. For this reason, very few biomarkers have been validated across laboratories, reflecting alterations in the nervous system in ALS. The most reproducible biomarker is the NFL, which can be used for diagnosis, prognosis, and disease monitoring in ALS (reviewed in [90]). Recent data show promising results for the diagnostic role of phosphorylated tau [78] and the prognostic role of GFAP [91,92], two structural proteins derived from neurons and astrocytes, respectively. Another promising biomarker is the cardiac troponin T, a protein of muscle origin, which was found to be increased in ALS patients [93] and reviewed in [83], demonstrating a role in disease progression monitoring.

Concerning TDP-43, some studies demonstrated an increased level of this protein in plasma and CSF of ALS patients, measured mainly through ELISA; however, the results vary across different studies, suggesting a low reproducibility of these assays, due to the low level of the pathological form in biofluids and the presence of immunoglobulins and albumins that can influence the binding of the antibody to the target TDP-43, as reviewed in [94,95]. Simoa^®^ technology, a new sensitive technology developed by Quanterix, has been developed and used for detecting TDP-43 in CSF and blood of ALS patients with a higher sensitivity than ELISA. However, the results are contradictory [96,97,98,99]. The most studied blood biomarkers are reported in Table 1.

Urine

Urine is an amber-colored biofluid produced by the renal system to expel excess liquids and waste products from the body. The kidney has the important role of regulating plasma osmolarity, adjusting the amount of water, electrolytes, and solutes in the blood circulation. The functional unit of the kidney is the nephron, where the glomerulus filters the plasma from the blood flow, and then through the renal tubule, the essential nutrients for the body are reabsorbed [100]. In a day, approximately 180 L of fluid is filtered, carrying out toxins, metabolic waste products, ions, and electrolytes to be eliminated. Specifically, urine is composed mainly of 95% water, 2% urea, 0.1% creatinine, 0.03% uric acid, and smaller amounts of metabolites, proteins, and several ions, such as chloride, sodium, potassium, sulfate, ammonium, and phosphate [101]. Only 20–30% of the blood proteins are also present in urine. Total urinary protein concentration is generally low, being below 0.2 mg per milliliter. This is because the renal system efficiently reabsorbs most proteins in circulation. Indeed, the Tamm–Horsfall protein (uromodulin), a kidney-specific glycoprotein, is the most abundant protein present in urine [102], whereas albumin and other blood proteins have a low concentration.

Urine has many advantages in the context of biomarker research. Indeed, it is easily accessible, and large quantities can be sampled, making it extremely feasible for repeated longitudinal measurements. Moreover, it may be a favorable biofluid for measuring low-abundance proteins, considering the low levels of blood-abundant proteins, which may affect the specificity and sensitivity of analytical assays. On the other hand, it has to be taken into account that proteins can cross the filtration barrier depending on their size and charge; i.e., proteins with a mass of 60–70 kDa as well as negatively charged proteins are largely retained in the capillary lumen and thus re-enter the systemic circulation [103], which may narrow their applicability for biomarker detection. It is also important to consider that urine composition is affected by several variables, such as gender, age, weight, pH, time of collection, handling, and diet [104,105,106].

The most studied urinary biomarker for ALS is neurotrophin receptor p75 extracellular domain (p75ECD), the detection of which indicates motor neuron degeneration, which correlates with disease progression [107,108]. Other candidate prognostic biomarkers have also been proposed, such as titin and collagen type IV, as reviewed in [108]. The most studied urine biomarkers are reported in Table 1.

Saliva

Saliva is the slightly acidic, hypotonic mucoserous biofluid secreted by major and minor salivary glands in the mouth. A human adult produces almost 0.5–1 L of saliva with a flow rate of 0.3 mL/min in unstimulated conditions and reaching a maximum of 7 mL/min after stimulation. Saliva secretion is regulated at the level of the salivary nuclei of the medulla, and specific stimuli, such as the act of chewing and gustatory stimuli, are known to induce hypersecretion. Particularly, saliva production is controlled by the sympathetic and parasympathetic nervous systems and several hormones [109].

Saliva comprises more than 99% water, in which electrolytes, immunoglobulin A, proteins, enzymes, mucin, urea, and ammonia are dissolved. The pH ranges between 6 and 7, depending on the secretion flow. The parotid saliva is enriched in amylase, proline-rich proteins, and agglutinins. Mucins MG1 and MG2 are mainly secreted by sublingual saliva and lysozyme; submandibular glands release Cystatin S [110,111].

Saliva has several roles, which include lubrication and protection of the mouth and the teeth, antibacterial activity, buffering action for the regulation of the pH, taste, and digestion [112,113]. This biofluid has gained attention in recent years as a non-invasive and easy-to-obtain source of biomarkers, but no saliva biomarkers have been associated with ALS yet. However, efforts are being made to identify saliva-associated biomarkers for ALS and other neurodegenerative diseases. For example, Carlomagno et al. employed Raman spectroscopy in a small cohort of 19 ALS patients, 10 PD patients, 10 AD patients, and 10 healthy controls. Particularly, they showed promising results in discriminating ALS from the spectra obtained, mainly due to differences in the concentration of lipids, but further investigation on larger cohorts is necessary [114]. Additionally, extracellular vesicles (EVs) derived from saliva are reported as being less abundant in contaminants, such as non-EV proteins or apolipoproteins, than plasma, which potentially enables an easy identification of novel biomarkers [115,116] (Table 1).

Tears

Tears are a transparent fluid produced by the lacrimal glands. They create a thin film that protects and lubricates the eyelids, conjunctiva, and cornea, subdivided into an outer lipid layer, a middle aqueous layer, and an inner glycocalyx layer [117]. Tears function not only by delivering nutrients to the cornea, but also by removing foreign material and protecting against infections.

Tears are aqueous solutions mainly made of water, containing electrolytes, proteins, lipids, mucins, defensins, collectins, and other small molecules [118]. The total protein concentration varies from 6 to 10 mg per milliliter of tears. Among the most abundant proteins are lactoferrin and lysozyme, both having antibacterial and antimicrobial functions, as well as lipocalin, which binds to the least soluble lipids, enhancing their solubility, and secretory IgA, which can react with antigens on bacterial cells [119,120,121].

Tear collection methods are minimally invasive, rapid, and painless, with glass microcapillary tubes or Schirmer strips being the cheapest and simplest tools used [122]. Tears can be used to study not only ocular surface diseases, but also systemic disorders [123]. On the other hand, the analysis is challenging due to the limited volume that can be collected, which is approximately 5 µL, highlighting the need for sensitive assays requiring low input volumes.

No tear biomarkers have been associated with ALS yet. Nevertheless, a recently published proteomic study found two proteins, namely SERPINC1 and HP, to be differentially present in ALS patients compared to controls [124]. Moreover, a metabolomic analysis of tear fluid collected from ALS patients highlighted a signature that differentiates bulbar and spinal forms of the disease, although no differences were reported between patients and controls [125] (Table 1).
cells-14-00848-t001_Table 1Table 1Most used and validated biomarkers for ALS in biofluids.BiofluidALS MarkersCSFNFL [126]neurofilament high (NFH and phosphorylated NFH) [80]tau and its phosphorylated form [82]myeloid protein chitotriosidase-1 (CHIT1) [127]glial protein YKL-40, also known as chitinase-3-like protein 1 (CHI3L1) [128]YKL-39 [79]inserting cryptic exons of the HDGFL2 [83]BloodNFL [79]phosphorylated tau [78]GFAP [91,92]cardiac troponin T [93]Urineneurotrophin receptor p75 extracellular domain (p75ECD) [79]Salivano saliva biomarkers have been associated with ALS yetTearsno tear biomarkers have been associated with ALS yet


## 3. Beyond Free Circulating Proteins: Extracellular Vesicles and Cell-Free DNA

Different biological entities, like extracellular vesicles (EVs) and cell-free DNA (cfDNA), are also present in biofluids and carry specific information about the site of origin. Technical hurdles and poor reproducibility across laboratories still hamper their use as biomarkers, but their potential must be considered in view of the technical challenges that are likely to be overcome in the future.

Extracellular vesicles (EVs)

EVs are nanoparticles released by all cells and consist of a phospholipidic bilayer that protects the cargo, comprising proteins, nucleic acids, and metabolites, from degradation by proteases, nucleases, and other enzymes present in circulation. EVs are usually classified into exosomes, microvesicles, and apoptotic bodies according to the biogenesis pathway from which they derive [129]. However, the biophysical characteristics, such as size and density, or the structural and molecular content, cannot fully distinguish the different subpopulations. Recently, circulating EVs have gained attention for their potential use as disease biomarkers. Different lines of investigation focus on unraveling the cargo of brain EVs in physiological and pathological conditions as a tool to study CNS alterations. The bidirectional passage of EVs between the CNS and the bloodstream has been reported, but not mechanistically explained. A few markers have been identified as specifically associated with brain-derived EVs and used to isolate these species from peripheral biofluids like blood. These markers include L1 cell adhesion molecule (L1CAM), Neural Cell Adhesion Molecule (NCAM), glutamate ionotropic receptor AMPA type subunits 2 and 3 (GluR2/3), GFAP, glutamate receptor, ionotropic, N-methyl D-aspartate 2A (NMDAR2A), ATPase, Na+/K+ transporting, and alpha 3 polypeptide (ATP1A3) [130,131,132,133,134,135]. However, it still remains poorly understood how EVs can cross barriers, particularly the BBB [136,137].

Several studies analyzed EV passage through the BBB in vitro. Fluorescently labeled human serum-derived EVs pass through immortalized murine-derived bEnd.3 cells [138], and HEK293T-derived EVs cross the human brain microvascular endothelial cells (BMECs) only upon stimulation with Tumor Necrosis Factor-alpha (TNF-α) [139] or Lipopolysaccharide (LPS) [140]. These two observations suggest that inflammatory signals may exert an important influence on BBB permeability.

Preliminary observations suggest clathrin-mediated endocytosis and micropinocytosis as cellular mechanisms favoring EV transcytosis to explain how EVs pass through the BBB. However, how the process is initiated per se remains the subject of open investigations [139,141]. More complex systems with microfluidics, BMECs, extracellular matrix, primary astrocytes, and pericytes confirmed the transcellular transport of EVs through endothelial cells mediated by endocytosis [141]. In vitro models and BMEC cells are suitable tools to study the interplay between EVs and the BBB; nevertheless, considering the complex multicellular structure of the BBB, they may not clearly explain the whole process of EV passage through the barrier.

Nevertheless, there is evidence for EV crossings from the periphery to the CNS and vice versa, although in an unbalanced manner. In physiological conditions, the passage of EVs from the periphery to the CNS is indeed an exceptional event, considering that most EVs accumulate in the liver, kidney, and spleen [140]. Injecting zebrafish embryos with fluorescently labeled peripheral EVs collected from AD patients’ serum resulted in EVs internalized by neurons and glial cells [138]. Through the injection of radioactively labeled EVs collected from human erythrocytes in the peripheral circulatory system of mice, EVs entered the microglia cells during LPS-induced systemic inflammation [140].

Another unresolved question regards EV uptake selectivity in vivo in the CNS. EVs derived from up to ten cell lines were radiolabeled and injected intravenously in CD-1 mice. All of them were retrieved in the brain, showing that all EV types crossed the BBB. A 10-fold variation rate was reported among cell lines, suggesting that specific surface proteins may favor EV uptake in the CNS [142]. Treatments with LPS, wheat germ agglutinin (WGA), and mannose 6-phosphate (M6P) also induce EV uptake by stimulating the adsorptive transcytosis [142]. Furthermore, the use of transgenic mice, in which the hematopoietic cells express a Cre protein that is sorted in EVs circulating in the bloodstream and recipient cells express a construct that is activated by the Cre protein leading to the irreversible expression of an enhanced yellow fluorescent protein, showed unequivocally that bloodstream EVs can be uptaken by synaptically active neurons [143]. Accordingly, intranasally delivered EVs accumulated in brain regions up to 96 h after administration [144]. On the other hand, regarding the passage of brain EVs into peripheral biofluids, radioactively labeled EVs injected into the ventricles passed from the brain to the bloodstream [142], also supporting the existence of EV flows from the brain into circulation. Consequently, brain-derived EVs can be isolated from the blood as potential CNS biomarkers [145,146]. In ALS and FTD patients, brain-derived EVs enriched with neuronal markers (L1CAM+) were isolated from plasma samples, and their cargo contained TDP-43 and tau proteins [99]. Remarkably, FTD individuals with a tau pathology presented a significant enrichment of tau protein in neuronal EVs; similarly, FTD (and ALS) patients with TDP-43 pathology had increased TDP-43 levels in neuronal EVs, suggesting the use of neuronal EVs to predict specifically tau or TDP-43 pathology associated with FTD subtypes [99]. Also, the microRNA content is noted to be altered in L1CAM-positive EVs in ALS patients [147,148,149]. Still, the lack of consensus in different laboratories on the specific RNAs that are deregulated in brain-derived EVs in disease conditions highlights the need to further validate this approach before applying it in clinical practice.

Circulating cell-free DNA (cfDNA)

In 1948, Mandel and Metais demonstrated for the first time the presence of circulating cell-free nucleic acids in healthy donors’ blood [150,151]. These nucleic acids comprise cell-free RNA (cfRNA) and DNA (cfDNA) and circulating cell-free mitochondrial DNA (cf-mtDNA) [152,153]. cfDNA is double-stranded and binds to histone proteins, and single nucleosomes, with a 160–180 bp size, can be detected in bioliquids upon release from apoptotic cells [152,153]. After the discovery of free nucleic acids in biological fluids like blood and CSF, cfDNA has received special attention as a potential biomarker, especially in cancer and prenatal research. In healthy individuals, cfDNA is derived from normal metabolic activity of hematopoietic cells, as shown by the methylome profile and in vivo generation of genome-wide nucleosome occupancy maps [154,155]. In general, plasma cfDNA content in healthy people is less than 10ng/mL, but it increases not only in several chronic and acute disorders (such as autoimmune diseases, cancer, diabetes, sepsis and transplantation), but also in several physiological conditions, such as pregnancy due to fetal DNA release, or during aging because of reduced clearance [155,156,157].

The use of cfDNA as a valuable diagnostic and prognostic biomarker came first from the cancer field [157,158,159,160]. In cancers located in the CNS, cfDNA has become an essential diagnostic and predictive biomarker, considering that the biopsy is risky or inaccessible [156]. In these cases, cfDNA analysis in the CSF proved to be more sensitive than standard cytological analysis of tumor cells, making it a promising new diagnostic biomarker [161]. The amount of CSF cfDNAs is also predictive, being higher in patients with advanced CNS tumors [162], and it is inversely correlated with the survival of glioma patients [163,164].

Another interesting application regards the analysis of the methylation profile of the cfDNA, which could be more informative than investigating specific disease-associated mutations, especially in neurodegenerative disorders, where the sporadic forms are more frequent than the familial ones. The advantage is that, by assessing the methylation profile, the cell of origin of the cfDNA is specifically determined [165]. For example, the methylation profile data of 8 ALS patients and 8 controls demonstrated a significant increase in the level of muscle-derived cfDNA in ALS patients’ blood compared to controls [166]. The methylation profile of plasma-derived cfDNA makes it possible to discriminate patients with AD from people with mild cognitive impairment (MCI) and healthy donors based on the amount of neuronal-derived cfDNA [167]. Furthermore, patients with severe traumatic brain injury and cardiac arrest with neuronal damage and BBB disruption also show elevated levels of brain-derived cfDNA in the bloodstream [157].

Circulating mitochondrial cfDNA (cf-mtDNA) is also gaining attention in neurodegenerative disease research, as it is released during cell stress and necrosis. Cf-mtDNA is usually detected by amplifying mitochondrial genes, such as COX3, using droplet digital PCR (ddPCR) [168]. Mitochondrial DNA is elevated in PD patients’ serum and MS CSF [169], while contradictory results have been reported [170] in AD. Changes in the cf-mtDNA amount were found in patients with CNS tumors and ALS [171].

Due to its high sensitivity, the analysis of cfDNA could serve as an early biomarker in liquid biopsy for various disorders.

By combining its analysis with other biomarkers, including EVs, the ability to diagnose diseases and predict their progression could greatly improve (Figure 3). In this perspective, Mugoni et al. have developed the ONCE protocol, which allows simultaneous RNA analysis from EVs and cfDNA from the same plasma aliquot [172] (Figure 3).

## 4. Conclusions

The tight structure of the barriers that protect the tissues of the central nervous system poses a challenge to researchers in understanding and targeting the CNS. For this reason, it is essential to determine their structure and how they function to implement strategies and novel routes for diagnosing, monitoring, and treating neurological disorders such as ALS. CNS-derived biomarkers are certainly useful in tracking changes in the cells of the nervous system. In ALS, biomarkers are actively investigated as promising tools to assess unmet needs, e.g., the effect of drug treatments. Several validated biomarkers worldwide, such as NFL, are currently measured in people with ALS, but some drugs may affect their clearance and yield confusing results [173,174]. Other problems that arise related to the candidate biomarkers identified are linked to the lack of reproducibility and technical challenges. Therefore, protein and nucleic acid biomarkers are being continuously investigated with an increasing interest towards brain-derived EVs and cell-free DNA from neuronal origin. In addition to understanding the neuronal origin of the molecular content of biofluids, deeper research should be conducted into the mechanisms that favor the mobility of molecules, proteins, and EVs via the CNS and intestinal barriers.

## Figures and Tables

**Figure 1 cells-14-00848-f001:**
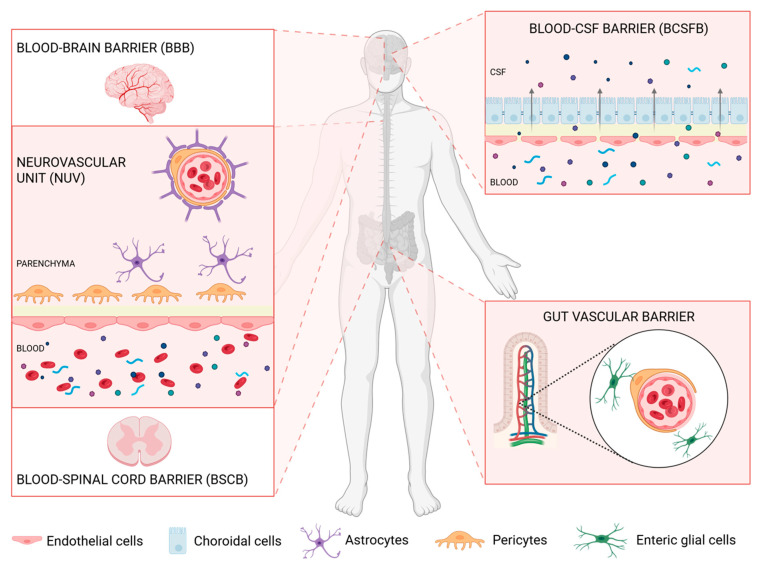
Representation of the main physical barriers controlling the passage of molecules and biological entities from and to the nervous system. The cell composition of the blood–brain barrier (BBB) and the blood–spinal cord barrier (BSCB) is illustrated on the left. On the right are the illustrations of the blood–CSF barrier (BSCFB) and the gut–vascular barrier (GVB). The illustration was realized with Biorender ©.

**Figure 2 cells-14-00848-f002:**
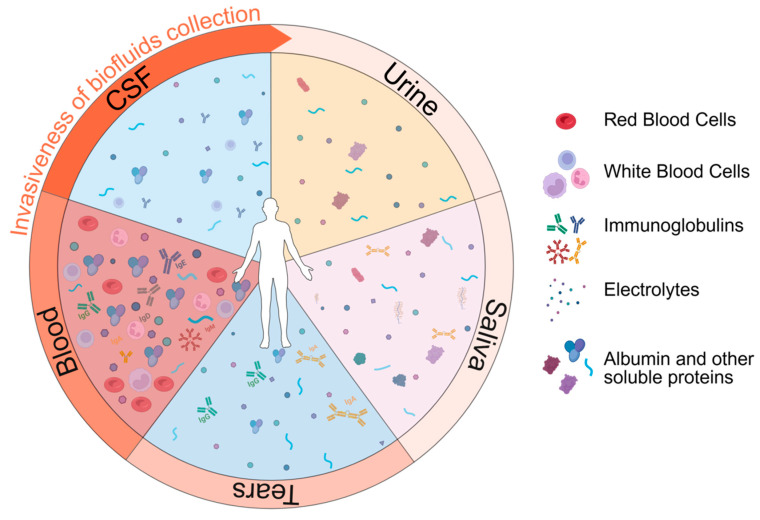
General representation of the biological complexity of the human biofluids (CSF, blood, urine, saliva, tears) used for biomarker discovery and monitoring. Illustration realized with Biorender©. Urine collection is generally considered the least invasive, while CSF collection is the most invasive.

**Figure 3 cells-14-00848-f003:**
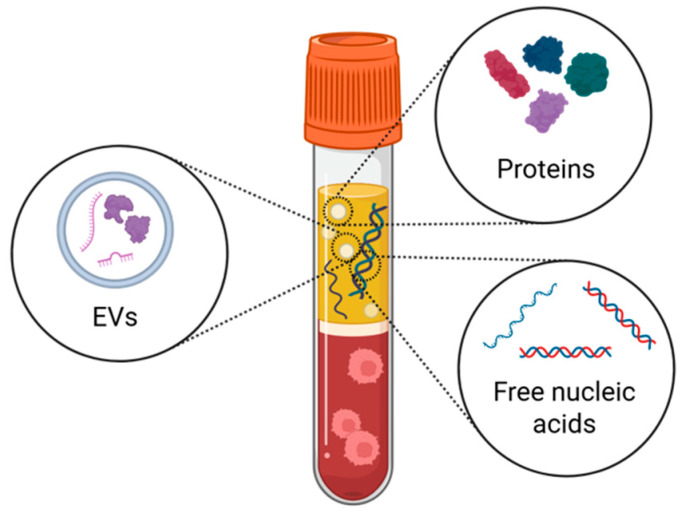
The potential of combining a single blood sample to analyze genomic material, proteins, extracellular vesicles, and metabolites. Illustration realized with Biorender ©.

## Data Availability

Not applicable.

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
