# Peer review of "Barriers in the Nervous System: Challenges and Opportunities for Novel Biomarkers in Amyotrophic Lateral Sclerosis"

_cells, 2025, doi:10.3390/cells14110848_

Round 1

Reviewer 1 Report

Comments and Suggestions for Authors

The authors present a comprehensive picture of the research conducted on ALS biomarkers. I appreciate the inclusion of several sections detailing the source of biomarkers and the type of barriers, which interface the nervous system with the rest of the organism. Figures are also well done and the table of protein biomarkers will be useful to consult for readers of this work.

I also liked the inclusion of EV and cDNAs as it is a rapidly expanding field. The potential pitfalls of these biomarkers connected to the difficulty of detection, the lack of reproducibility and the uncertainty as to the molecular mechanisms regarding their leakage from the brain tissue have bee appropriately mention.

For the reasons expressed above, I believe this review is timely and a fair representation of the state of the field. Thus, I am in favor of publication as is.

1.The authors present a comprehensive picture of the research conducted on ALS biomarkers. I appreciate the inclusion of several sections detailing the source of biomarkers and the type of barriers which interface the nervous system with the rest of the organism. I also liked the inclusion of EVs and cfDNAs as it is a rapidly expanding field. The potential pitfalls of these biomarkers connected to the difficulty of detection, the lack of reproducibility and the uncertainty as to the molecular mechanisms regarding their leakage from the brain tissue have been appropriately mentioned. I believe the review provides a fair and balanced overview of the field. Biomarker research for ALS meets and important clinical need, however we have very few biomarkers available at the moment. I believe more effort in identifying novel biomarkers is needed and thus this topic will be of interest for the community,

2. reviews on biomarkers are far and few in between compared to other topics. The last author published a similar review a few years ago. While the field has not changed that much, she tried to include mention of EVs and cfDNA, which are a less understood and more recent development in the field. I, thus, think that the review is timely and would be of general interest.

3. the references are appropriate. I did not think there was excessive auto-citation. The field seem to be fairly represented.

4.the information is well balanced and generally supported by the papers cited.

5. both figures and the table are well done and easy to read. The table of protein biomarkers will be useful to consult for readers of this work and for people to cite and use as a reference. 

Author Response

We thank the Reviewer for their careful reading of the manuscript and their encouraging comments. Below, we quote the original comments in italics and respond to each point in regular font.

Comment:

The authors present a comprehensive picture of the research conducted on ALS biomarkers. I appreciate the inclusion of several sections detailing the source of biomarkers and the type of barriers, which interface the nervous system with the rest of the organism. Figures are also well done, and the table of protein biomarkers will be useful to consult for readers of this work.

I also liked the inclusion of EVs and cDNAs, as it is a rapidly expanding field. The potential pitfalls of these biomarkers, connected to the difficulty of detection, the lack of reproducibility, and the uncertainty as to the molecular mechanisms regarding their leakage from the brain tissue, have been appropriately mentioned.

For the reasons expressed above, I believe this review is timely and a fair representation of the state of the field. Thus, I am in favor of publication as is.

  1. The authors present a comprehensive picture of the research conducted on ALS biomarkers. I appreciate the inclusion of several sections detailing the source of biomarkers and the type of barriers that interface the nervous system with the rest of the organism. I also liked the inclusion of EVs and cfDNAs, as it is a rapidly expanding field. The potential pitfalls of these biomarkers are connected to the difficulty of detection, the lack of reproducibility, and the uncertainty as to the molecular mechanisms regarding their leakage from the brain tissue have been appropriately mentioned. I believe the review provides a fair and balanced overview of the field. Biomarker research for ALS meets an important clinical need; however, we have very few biomarkers available at the moment. I believe more effort in identifying novel biomarkers is needed, and thus, this topic will be of interest for the community.
  2. Reviews on biomarkers are far and few in between compared to other topics. The last author published a similar review a few years ago. While the field has not changed that much, she tried to include mention of EVs and cfDNA, which are less understood and more recent developments in the field. I, thus, think that the review is timely and would be of general interest.
  3. The references are appropriate. I did not think there was excessive auto-citation. The field seems to be fairly represented.
  4. The information is well balanced and generally supported by the papers cited.
  5. Both the figures and the table are well done and easy to read. The table of protein biomarkers will be useful to consult for readers of this work and for people to cite and use as a reference.

Response:

We thank the reviewer for the positive comments. We appreciate that the Figures and Tables are comprehensive and help readers follow the flow of the review.

Reviewer 2 Report

Comments and Suggestions for Authors

This review addresses a highly relevant and timely topic in the field of neurodegeneration, specifically focusing on the ongoing challenges and emerging opportunities in the identification of biomarkers for Amyotrophic Lateral Sclerosis (ALS). The manuscript provides a clear overview of the biological and physical barriers to biomarker discovery in the central nervous system, and appropriately emphasizes the potential of peripheral biofluids for non-invasive assessment. Notably, I think that the discussion of established biomarkers such as neurofilament light chain (NfL), alongside emerging candidates like extracellular vesicles (EVs) and cell-free DNA, is particularly informative and effectively reflects the current state of research in the field.

Major points

The value of this manuscript could be further enhanced if the authors clearly classify the biomarkers according to their intended clinical purpose—namely, diagnosis (early detection of ALS), prognosis (prediction of disease outcome or survival), or disease progression monitoring (tracking changes over time). I encourage the authors to reorganize the discussion with this perspective in mind.

Minor points

  1. Missing Reference Numbers:

There are missing or skipped reference numbers in the manuscript. Some citations appear out of sequence, or the corresponding references cannot be found in the reference list. Accurate citation is essential for ensuring the credibility of the work, so I recommend carefully reviewing the reference numbering and ensuring consistency between the in-text citations and the reference list.

  1. Overly Fragmented Paragraph Structure:

In some sections, the paragraph structure is overly fragmented, which disrupts the logical flow and coherence of the discussion. Merging closely related content into more cohesive paragraphs would improve the readability and clarity of the manuscript. In particular, the sections discussing individual biomarkers could benefit from consolidation and clearer organization.

Author Response

We thank the Reviewer for their careful reading of the manuscript and their constructive suggestions. Below, we quote the original comments in italics and respond to each point in regular font.

Comment:

This review addresses a highly relevant and timely topic in the field of neurodegeneration, specifically focusing on the ongoing challenges and emerging opportunities in the identification of biomarkers for Amyotrophic Lateral Sclerosis (ALS). The manuscript provides a clear overview of the biological and physical barriers to biomarker discovery in the central nervous system, and appropriately emphasizes the potential of peripheral biofluids for non-invasive assessment. Notably, I think that the discussion of established biomarkers such as neurofilament light chain (NfL), alongside emerging candidates like extracellular vesicles (EVs) and cell-free DNA, is particularly informative and effectively reflects the current state of research in the field.

Major points

The value of this manuscript could be further enhanced if the authors clearly classify the biomarkers according to their intended clinical purpose—namely, diagnosis (early detection of ALS), prognosis (prediction of disease outcome or survival), or disease progression monitoring (tracking changes over time). I encourage the authors to reorganize the discussion with this perspective in mind.

Response:

We thank the reviewer for the suggestions. We reorganized the discussion of the different biomarkers in each biofluid, specifying their potential utility in clinical practice. We hope that the new version is more informative. Please find below the major changes highlighted in bold.

This is an example regarding the CSF:

‘The most reproducible biomarkers for ALS that have been identified in the CSF are neuronal structural proteins such as NFL, neurofilament heavy (NFH and phosphorylated NFH), tau, and its phosphorylated forms (phosphorylated Tau 181 and 217) [80]. In addition, there are also chitinases, enzymes secreted by macrophage and activated microglia in the inflammatory state, namely the myeloid protein chitotriosidase-1 (CHIT1), the glial protein YKL-40, also known as chitinase-3-like protein 1 (CHI3L1), and YKL-39, also known as chitinase-3-like protein 2 (CHI3L2) (reviewed in [81]). Specifically, each of these markers has different applications in clinical practice. Neurofilaments have been identified as the most promising biomarkers for ALS, having diagnostic, prognostic, and disease monitoring roles [82]. Chitinases demonstrate diagnostic value, with CHIT1 and CHIT3L1 being associated with disease severity and progression [83]. Instead, tau and its phosphorylated forms correlate with disease severity [84]. Of interest, a novel diagnostic protein biomarker is derived from the cryptic exon-containing hepatoma-derived growth factor-like protein 2 (HDGFL2), whose content in the CSF is strictly dependent on TDP-43 loss of splicing activity in ALS and FTD and is significantly increased in sporadic ALS patients as well as in presymptomatic and symptomatic C9orf72 mutation carriers [85]. Unfortunately, the CSF collection is an invasive intervention with a lumbar puncture [86]. Therefore, the applicability of these biomarkers is limited and not for routine monitoring.’

Here the blood:

‘The most reproducible biomarker is the NFL, which can be used for diagnosis, prognosis, and disease monitoring in ALS(reviewed in [92]). Recent data show promising results for the diagnostic role of phosphorylated tau [80] and the prognostic role of GFAP [93,94], two structural proteins derived from neurons and astrocytes, respectively. Another promising biomarker is the cardiac troponin T, a protein of muscle origin, which was found to be increased in ALS patients [95] and reviewed in [85], demonstrating a role in disease progression monitoring.’

Finally, the urines:

‘The most studied urinary biomarker for ALS is neurotrophin receptor p75 extracellular domain (p75ECD), the detection of which indicates motor neuron degeneration, which correlates with disease progression [109]. Other candidate prognostic biomarkers have also been proposed, such as titin and collagen type IV, as reviewed in [109]. 

Comment:

Minor points

  1. Missing Reference Numbers:

There are missing or skipped reference numbers in the manuscript. Some citations appear out of sequence, or the corresponding references cannot be found in the reference list. Accurate citation is essential for ensuring the credibility of the work, so I recommend carefully reviewing the reference numbering and ensuring consistency between the in-text citations and the reference list.

Response:

We are sorry for the misalignment in the bibliography and missing references. We thank the reviewer for the comment. We have adjusted the references and the corresponding numbers to ensure an accurate citation. 

Comment:

Overly Fragmented Paragraph Structure:

In some sections, the paragraph structure is overly fragmented, which disrupts the logical flow and coherence of the discussion. Merging closely related content into more cohesive paragraphs would improve the readability and clarity of the manuscript. In particular, the sections discussing individual biomarkers could benefit from consolidation and clearer organization.

Response:

We appreciate the suggestion of the reviewer, and we have revised the paragraphs and reorganized the sentences to provide better readability and flow. We thank the Reviewer for this suggestion.